# Combining Radiation- with Immunotherapy in Prostate Cancer: Influence of Radiation on T Cells

**DOI:** 10.3390/ijms23147922

**Published:** 2022-07-18

**Authors:** Diana Lindner, Claudia Arndt, Liliana Rodrigues Loureiro, Anja Feldmann, Alexandra Kegler, Stefanie Koristka, Nicole Berndt, Nicola Mitwasi, Ralf Bergmann, Marcus Frenz, Michael P. Bachmann

**Affiliations:** 1Helmholtz-Zentrum Dresden-Rossendorf, Institute of Radiopharmaceutical Cancer Research, 01328 Dresden, Germany; diana@mlindner.eu (D.L.); c.arndt@hzdr.de (C.A.); l.loureiro@hzdr.de (L.R.L.); a.feldmann@hzdr.de (A.F.); a.kegler@hzdr.de (A.K.); s.koristka@gmx.de (S.K.); n.berndt@hzdr.de (N.B.); n.mitwasi@hzdr.de (N.M.); r.bergmann@hzdr.de (R.B.); 2Tumor Immunology, University Hospital Carl Gustav Carus, University Cancer Center (UCC), Technical University Dresden, 01307 Dresden, Germany; 3Institute of Biophysics and Radiation Biology, Semmelweis University, 1094 Budapest, Hungary; 4Faculty Informatik and Wirtschaftsinformatik, Provadis School of International Management and Technology AG, 65926 Frankfurt, Germany; marcus.frenz@doz-provadis-hochschule.de; 5National Center for Tumor Diseases (NCT), Partner Site Dresden, 01307 Dresden, Germany; 6German Cancer Consortium (DKTK), Partner Site Dresden and German Cancer Research Center (DKFZ), 69120 Heidelberg, Germany

**Keywords:** prostate stem cell antigen, prostate cancer, radiation, immunotherapy, bispecific T cell engager

## Abstract

Radiation of tumor cells can lead to the selection and outgrowth of tumor escape variants. As radioresistant tumor cells are still sensitive to retargeting of T cells, it appears promising to combine radio- with immunotherapy keeping in mind that the radiation of tumors favors the local conditions for immunotherapy. However, radiation of solid tumors will not only hit the tumor cells but also the infiltrated immune cells. Therefore, we wanted to learn how radiation influences the functionality of T cells with respect to retargeting to tumor cells via a conventional bispecific T cell engager (BiTE) and our previously described modular BiTE format UNImAb. T cells were irradiated between 2 and 50 Gy. Low dose radiation of T cells up to about 20 Gy caused an increased release of the cytokines IL-2, TNF and interferon-γ and an improved capability to kill target cells. Although radiation with 50 Gy strongly reduced the function of the T cells, it did not completely abrogate the functionality of the T cells.

## 1. Introduction

Over the past decades diverse strategies were developed for retargeting of immune cells to tumor cells using conventional antibodies (Abs) or recombinant derivatives e.g., bispecific antibodies (bsAbs) or immune cells genetically modified to express Chimeric Antigen Receptors (CARs) (e.g., [1,2,3,4,5,6,7,8,9,10]). While these humoral and cellular immunotherapy options are highly efficient in B-cell leukemias, their application is limited for the treatment of solid tumors which may be due to mechanical barriers, the lack of expression of suitable homing receptors and adhesion molecules which interferes with an efficient infiltration of immune cells into tumor tissues. Moreover, the immunosuppressive tumor microenvironment including the expression of check point inhibitors leads to a downregulation of infiltrated immune effector cells (e.g., [11,12,13]).

Unfortunately, the treatment of patients with cytostatic drugs or radiation causes a selection pressure on the tumor cells favoring the outgrowth of resistant escape variants (e.g., [14,15,16,17]). As shown recently by us, radioresistant tumor cells can still be recognized and destroyed by retargeting of T cells [18,19]. Thus, a combination of radiation therapy with T cell-based immunotherapy appears to be logical and feasible especially when bearing in mind that radiotherapy can increase the local expression of multiple cytokines including for example interferons as well as homing factors such as the chemokines CXCL9-11, CXCL10, and CXCL11. In addition, radiation may also affect the tumor blood vessels, and cause an upregulation of adhesion molecules thereby favoring the invasion of immune cells into the tumor tissue and improving the local conditions of the tumor microenvironment for immunotherapy (e.g., [20,21]).

Obviously, during radiotherapy, not only the tumor cells will be irradiated but also the immune cells present in the tumor tissue. So far, little is known how beam- or endoradionuclide associated radiation affects the functionality and especially the killing capability of tumor infiltrating immune effector cells. Therefore, we decided to analyze the effect of radiation on the functionality of immune effector T cells during retargeting of tumor cells.

In 2014 we showed that conventional bispecific T cell engagers (BiTEs) can be replaced by a highly flexible universal modular bispecific antibody platform (UNImAb, see also Figure 1) [22,23]. The UNImAb system consists of two antibody components: (i) one module, the so-called target module (TM), is a bifunctional molecule which on the one hand recognizes and binds to the target cell via e.g., an Ab domain, and on the other hand is fused to a peptide epitope tag; (ii) the second component is a bsAb which we termed effector module (EM). For retargeting of T cells one of the two Ab domains of the EM is directed to the CD3 complex of the T cell receptor complex while the other arm is directed to the peptide epitope tag of the TM. Consequently, the EM and TM can form a bispecific immune complex which binds on the one hand to the T cell via the CD3 Ab domain of the EM and on the other hand to the target cell via the Ab domain of the TM. Thus, the UNImAb complex behaves similarly to a conventional BiTE. Similarly to a conventional BiTE, the bispecific UNImAb complex can form a cross linkage between the effector T cell and the target cell which leads to an activation of the engaged T cell and finally to the killing of the target cell. As shown in our previous studies there is no difference with respect to the killing capability of T cells whether they are cross-linked via a UNImAb complex or a conventional BiTE [22,23]. However, the modular BiTE format UNImAb has several advantages: one can easily alter the targeting specificity of the UNImAb complex simply by replacing the respective TM. In contrast to the construction of a novel BiTE the resulting UNImAb complex will be functional without intense optimization [7,24]. Moreover, several TMs can be combined with the same EM and applied in parallel, thereby allowing simultaneous targeting of multiple tumor targets also known as combinatorial or gated targeting (OR or AND gated targeting) [25,26,27]. Furthermore, as the peptide/anti-peptide domain used in the UNImAb system is the same as in our adapter CAR system UniCAR (e.g., [7,8,9,10]) which is currently being tested in clinical phase 1 trials (NCT04633148, NCT04230265), the same TM can not only be used for retargeting of unmodified T cells via the modular BiTE platform UNImAb but also via genetically modified UniCAR T or NK cells.

In the current proof of concept study, we analyzed the retargeting of prostate cancer (PCa) cells expressing as target the prostate stem cell antigen (PSCA) via a conventional anti-CD3/anti-PSCA BiTE (e.g., [3]) with the corresponding UNImAb format [22,23] and tested how radiation may affect the functionality of the redirected immune effector T cells. According to the here presented data, radiation conditions of a patient with PCa [28] would not impair the retargeting and especially not the killing capability of redirected effector T cells but should modulate the tumor microenvironment in favor of a T cell based immunotherapy.

## 2. Results

### 2.1. Conventional Bispecific T Cell Engager versus the Modular UNImAb System

As schematically summarized in Figure 1, a conventional bispecific T cell engager (BiTE) consists of two antibody (Ab) domains directed on the one hand to the CD3 complex of the T cell (Figure 1, Conventional BiTE, anti-CD3) and on the other hand to a tumor associated antigen (TAA) (Figure 1, Conventional BiTE, anti-TAA) present on the surface of the target cell. Each Ab domain consists of the respective variable heavy and light chain portions of an Ab which are usually fused via a glycine serine linker to form a single chain fragment variable (scFv). Both scFvs are combined e.g., in a tandem orientation via an additional glycine serine linker which results in the bispecific antibody (bsAb) (Figure 1, Conventional BiTE, bsAb). The modular BiTE format UNImAb consists of two components a universal effector module (EM) (Figure 1, Modular BiTE format, effector module) and a tumor specific target module (Figure 1, Modular BiTE format, target module). Incubation of the TM with the EM results in an immune complex via the peptide epitope tag of the TM and the anti-peptide scFv of the EM (Figure 1, Modular BiTE format). The formed immune complex behaves similarly to a conventional BiTE and can cross-link the immune cell via its anti-CD3 domain with the target cell via its anti-TAA domain.

### 2.2. Retargeting of Tumor Cells with Irradiated T Cells

#### 2.2.1. Estimation of Lysis Capability

Irradiated and unirradiated pan T cells were prepared and used for retargeting to PSCA overexpressing PC3 cells either in the absence (Figure 2, w/o) or presence of 30 pmol/mL of the respective Ab components (Figure 2, effector module alone (EM), target module (TM) alone, UNImAb complex (UNImAb), conventional BiTE (BiTE)). Incubation was performed for either 24 or 48 h. Using standard ^51^Cr release assays, the specific tumor cell lysis was determined for three independent donors. In agreement with our previously collected data the UNImAb format performed comparably well as the conventional BiTE format (Figure 2). Most importantly, we did not observe a dramatic loss of killing capability up to an irradiation of 20 Gy. After an exposure to 30 Gy the killing was reduced to a minor extend. But even after radiation with 40 or 50 Gy the T cells were still able to kill the target cells although the killing efficacy was clearly reduced.

#### 2.2.2. Estimation of T Cell Activation

Pan T cells were irradiated with up to 20 Gy and redirected to PC3-PSCA cells in the absence (Figure 3, w/o) or presence of 30 pmol/mL of the respective Ab components (Figure 3, effector module alone (EM), target module (TM) alone, UNImAb complex (UNImAb), conventional BiTE (BiTE)). Incubation was performed for 5 d. Afterwards the T cells were stained for expression of CD25, which was estimated by FACS analysis. As shown in Figure 3, both the conventional BiTE as well as the UNImAb format results in a comparable upregulation of the activation marker CD25.

#### 2.2.3. Estimation of Cytokine Release

Pan T cells were irradiated with up to 50 Gy and redirected to PC3-PSCA cells in the absence (Figure 4, Figure 5 and Figure 6, w/o) or presence of 30 pmol/mL of the respective Ab components (Figure 4, Figure 5 and Figure 6, effector module alone (EM), target module alone (TM), UNImAb complex (UNImAb), conventional BiTE (BiTE)). Incubation was performed for 24 h. Afterwards the concentrations of pro-inflammatory cytokines (IFN-γ (Figure 4), IL-2 (Figure 5), and TNF (Figure 6) were measured in the supernatants via ELISA. As the absolute cytokine concentrations of individual donors strongly varied, we show separately the mean cytokine concentrations ± SD of triplicates for three different donors (Figure 4, Figure 5 and Figure 6). In general, and for all the cytokines measured we observed a two to three-fold increased release of cytokines after irradiation with a peak mostly in the range between 6 and 10 Gy. Cytokine levels usually decrease below the levels of untreated T cells in case the T cells are irradiated with 30 Gy or more.

#### 2.2.4. Estimation of Proliferation

Pan T cells were irradiated with up to 20 Gy and redirected to PC3-PSCA cells in the absence (Figure 7, w/o) or presence of 30 pmol/mL of the respective Ab components (Figure 7, effector module alone (EM), target module alone (TM), UNImAb complex (UNImAb), conventional BiTE (BiTE)). Incubation was performed for 5 d. Prior to the incubation the T cells were stained with the proliferation dye eFluor^®^ 670 and proliferation was measured by FACS analysis. Redirection of pan T cells via the conventional CD3-PSCA BiTE or the UNImAb format leads to a significant proliferation of T cells up to 6 Gy radiation doses.

#### 2.2.5. Estimation of Expansion

Pan T cells were irradiated with up to 50 Gy and redirected to PC3-PSCA cells in the absence (Figure 8, w/o) or presence of 30 pmol/mL of the respective Ab components (Figure 8, effector module alone (EM), target module (TM) alone, UNImAb complex (UNImAb), conventional BiTE (BiTE)). Incubation was performed for 5 d. The absolute numbers of T cells were counted using the MACSQuant Analyzer and the ratio (t_1_/t_0_) was calculated. As shown in Figure 8, redirection of pan T cells via the anti-CD3-PSCA BiTE or the UNImAb format results in a comparable expansion rate of unirradiated T cells. However, already the low radiation dose of 2 Gy leads to an impairment of T cell expansion.

## 3. Discussion

Prostate cancer (PCa) is still the world’s second most common cancer in men [29]. Fortunately, around 80% of patients are diagnosed with a localized tumor [30,31]. Treatment options for localized disease are radical prostatectomy, external beam radiotherapy (EBRT), and brachytherapy. Unfortunately, relapses are not uncommon [32], especially in patients with high-risk disease (Prostate specific antigen (PSA) > 20 ng/mL, Gleason score > 8, clinical T stage of at least cT2c) [33]. For patients with metastatic disease the situation is even worse: Treatments are no more curative but only palliative in order to slow down the course of disease. Although the treatment with next generation androgen inhibitors, chemotherapy with taxanes, poly (ADP-ribose) polymerase (PARP) inhibitors and endoradionuclide therapy can delay disease, resistance to these treatments seems to be inevitable and thus most patients diagnosed with metastatic disease will finally die of PCa [34].

While the treatment of several cancers with immune checkpoint inhibitors (ICI) have shown impressive outcomes [35,36,37] in case of PCa only a small number of patients show a favorable response [38,39]. Interestingly, in few cases radiation therapy (RT) caused a tumor regression outside the irradiated field suggesting that RT can induce a systemic immunological anti-tumor response [40,41]. Unfortunately, the combination of hypofractionated RT with ICI did not turn in an increased overall survival (OV) [42] leading to the question how much radiation the local immune players can tolerate. For example, low dose radiotherapy (LDRT) with doses between 0.5 to 2 Gy can lead to an upregulation of cytokines and chemokines as well as an increase in infiltrating immune cells [33,43,44]. Consequently, radiation of a tumor should lead to an alteration of the immunosuppressive tumor microenvironment (TME) which supports the idea of conversion of an immunological “cold” tumor into a hot tumor that can be recognized by the immune system.

The modulation of the TME in this sense should also directly help to allow the retargeting of T cells via bispecific antibodies (bsAbs) or in form of CAR T cells of a cold tumor such as PCa. Both targeting strategies are based on the formation of synapses between the T cell and the target cell [45,46]. The interaction leads to the activation of the immune cell, release of pro-inflammatory cytokines and finally the killing of the target cell (e.g., [1,2,3,4,5,6,7,8,9,10] introduction). Therefore, we wanted to learn how radiation may influence the capability of redirected immune cells. In order to reproduce the data, we compared the results obtained for a conventional BiTE which is currently in a clinical phase 1 trial (NCT03927573) with our previously described modular BiTE format (UNImAb) (e.g., [22,23]). Both T cell engaging approaches are targeting prostate stem cell antigen (PSCA) which is overexpressed and thus a promising target on PCa cells [47,48].

Our data show that low radiation doses between 2 and 4 Gy as they are used in fractionated radiation schemes do not harm the killing capability of effector T cells. In contrast, at low doses of irradiation we see an even enhanced lysis of tumor cells, although the T lymphocytes appear to be quite sensitive to radiation. In spite of proliferation, we see no overall expansion. According to these data, only a portion of the radiated T cells proliferate while another portion of them is dying soon after radiation. The overall loss of T cells is obviously not compensated by proliferation. In order to explain the improved lysis capability, the T cells remaining after radiation should have an enhanced killing capability. Indeed, radiation up to 20 Gy resulted not only in an activation of the immune cells as indicated by the upregulation of CD25, an increased secretion of pro-inflammatory and T cell maintaining cytokines including IFN-γ, TNF and IL-2 but also in an enhancement of their ability to kill their target cells. Bearing in mind that Tregs express high affinity IL-2 receptors the increased IL-2 levels obviously are not favoring the function of the immunosuppressive Treg cells present in the pan T cell preparation. One possible explanation may be that the Tregs are more sensitive to irradiation. Together with the released pro-inflammatory cytokines this may be an additional reason for the improved functionality of the effector T cells. Besides fractionated radiation schemes also single radiation doses of up to 60 Gy are clinically used. Interestingly, even when applying such high doses of radiation, a portion of the radiated T cells still remained functional and were even able to proliferate. Taken together, the majority of radiated T cells are going to die but there will be sufficient time for them to attack and kill tumor cells. Moreover, their local activation will convert the cold tumor into a hot one, thereby attracting additional non-irradiated T cells into the tumor tissue. Furthermore, when solid tumors are irradiated there might be a radiation gradient in the tumor and the surrounding healthy tissue. Frequently, T cells are found surrounding solid tumors. These T cells may in part be exposed to favoring low doses which may help to activate them and cause them to enter into the tumor.

In summary, our data indicate that especially low dose of radiation should modulate the TME in favor of a redirection of immune cells but not impair their retargeting capability, thus supporting a combination of RT with targeted immunotherapy. According to our data, one would expect that the immunotherapy should be started in parallel to the RT. An advantage of the combination with RT should not only be true for retargeting via BiTEs but also CAR T cells. We favor the use of modular BiTE and CAR platforms such as the UNImAb or the UniCAR system because their targeting modules can be used as theranostic compound including for imaging, local endoradionuclide therapy, costimulation [49], and immunotherapy [50].

## 4. Materials and Methods

### 4.1. Cell Lines

The PCa cell line PC3 was genetically modified to overexpress human PSCA as described previously (e.g., [51]). The PC3-PSCA cell line was cultured in RPMI complete medium. All antibodies were expressed using permanently transduced mouse 3T3 cell lines. Mouse 3T3 cells were cultured in DMEM complete media (e.g., [52]). All cell lines were incubated at 37 °C with 5% CO_2_.

### 4.2. Isolation of PBMCs and T Cells

Buffy coats, supplied by the German Red Cross (Dresden, Germany), were used for the isolation of human PBMCs with the consent of the donors. The isolation of PBMCs from buffy coats and the following isolation and cultivation of untouched pan T cells were performed as described previously (e.g., [53]).

### 4.3. Expression and Purification of Antibodies

The construction of all recombinant antibodies (Abs) including the effector module (EM), the anti-PSCA target module (TM) and the anti-CD3-PSCA bsAb has been published previously (e.g., [3,51,52]). The EM is directed on the one hand to the CD3 complex of T cells on the other hand to the peptide epitope (E5B9) derived from the nuclear autoantigen La/SS-B (e.g., [3,54,55]). The TM consists of the same anti-PSCA domain as used in the anti-CD3-PSCA bsAb and represents a fusion protein of the anti-PSCA scFv domain with the E5B9 peptide epitope. All Ab derivates were isolated from cell culture supernatants of permanently transduced mouse 3T3 cells. All Ab constructs contain a C-terminal 6xhistidine tag and were purified from the supernatants using Ni-NTA affinity chromatography. Purified proteins were analyzed by SDS-PAGE and Western Blot as described previously.

### 4.4. Estimation of Cell Numbers

The cell numbers were determined via the MACSQuant Analyzer (Miltenyi Biotec GmbH, Bergisch Gladbach, Germany) using propidium iodide (PI) to assess cell viability and exclude non-viable cells.

### 4.5. Irradiation of T Cells

7.5 × 10^5^ human T cells were transferred in RPMI 1640 medium into a 2 mL reaction tube (Sarstedt, Nümbrecht, Germany). Each reaction tube was treated with a single dose in the range from 2 to 50 Gy as indicated in the respective Figures using the Gammacell^®^ 3000 Elan (Nordion International Inc., Ottawa, ON, Canada).

### 4.6. Cytotoxicity Assay

T cell mediated cytotoxicity was measured using Chromium-51 (^51^Cr) release assays. Radiolabeling of target cells was performed as described previously (e.g., [49]). 5 × 10^3 51^Cr labeled PC3-PSCA cells were pipetted to each well of a 96-well round bottom plate. 2.5 × 10^4^ pan T cells were added (equivalent to an E:T ratio of 5:1). The cell mixtures were incubated either in the absence of any Ab construct or the presence of 30 pmol/mL of the CD3-PSCA BiTE, the EM alone, the TM alone or the complex of the TM and EM. Complete RPMI medium was added so that each well contained a total volume of 200 µL. Afterwards the plate was incubated for 24 and 48 h. All samples were measured as triplets. The measurement and calculation of the specific lysis was conducted as described previously (e.g., [51]).

### 4.7. Activation Assay

5 × 10^3^ PC3-PSCA cells were incubated with 2.5 × 10^4^ pan T cells in the absence or presence of 30 pmol/mL of the anti-CD3-PSCA bsAb or the EM alone or the TM alone or the complex of the TM and EM. Incubation was performed for 5 d in 200 µL complete RPMI medium. All samples were measured as triplets using a 96-well round bottom plate. The plates were centrifuged and the supernatants were separated. T cells of one triplet were pooled, stained with anti-CD25/PE and CD3/PE-Vio770TM mAbs (Miltenyi Biotec GmbH) and measured by flow cytometry analysis using the MACSQuant Analyzer (Miltenyi Biotec GmbH).

### 4.8. Estimation of Proliferation and Expansion

Before radiation, T cells were stained with the proliferation dye eFluor^®^ 670 (Thermofisher Scientific, Dreieich, Germany). After 5 d of incubation, under the same conditions as mentioned above, the proportion of eFluor^®^ 670 diminished T cells was determined by flow cytometry using the MACSQuant Analyzer (Miltenyi Biotec GmbH). All samples were measured as triplets and T cells of one triplet were pooled.

In order to analyze T cell expansion, T cells were counted before (t_0_) and after 5 d of incubation (t_1_) by flow cytometry. All samples were measured as triplets. The ratio n(t_1_):n(t_0_) of each sample was calculated and the arithmetic mean was determined for each triplet.

### 4.9. Cytokine ELISA

The release of IL-2, IFN-γ or TNF by T cells was measured after 24 h of incubation by ELISA using the OptEIA Human IL-2, IFN-γ and TNF ELISA Kits (BD Biosciences, Heidelberg, Germany).

### 4.10. Statistics

Data were plotted and analyzed using the GraphPad Prism software 9.0 (La Jolla, CA, USA). Statistical analysis was performed by one-way ANOVA and Bonferroni Multiple Comparison test. The *p* values < 0.05 were considered significant (* *p* < 0.05; ** *p* < 0.01 and *** *p* < 0.001).

## 5. Conclusions

Pan T cells were redirected to PSCA expressing prostate cancer cells via a conventional anti-PSCA BiTE and a modular BiTE format (UNImAb). Both retargeting strategies performed equally well. Radiation of the T cells prior to the lysis studies applying clinically used radiation doses in the range from 2 to 50 Gy, which do not destroy their capability to eliminate the target cells. Radiation of up to around 10 to 20 Gy even improve the lysis of the target cells, most likely due to an increased release of cytokines including of TNF, IFN-γ, and IL-2. The presented data support the combination of radiation therapy with cellular immunotherapies.

## Figures and Tables

**Figure 1 ijms-23-07922-f001:**
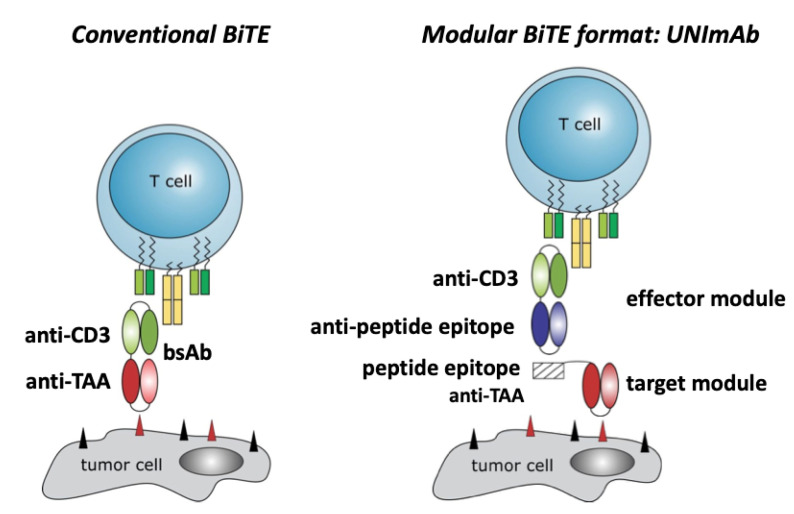
Schematic comparison of the conventional (**left**) and modular BiTE format: UNImAb (**right**).

**Figure 2 ijms-23-07922-f002:**
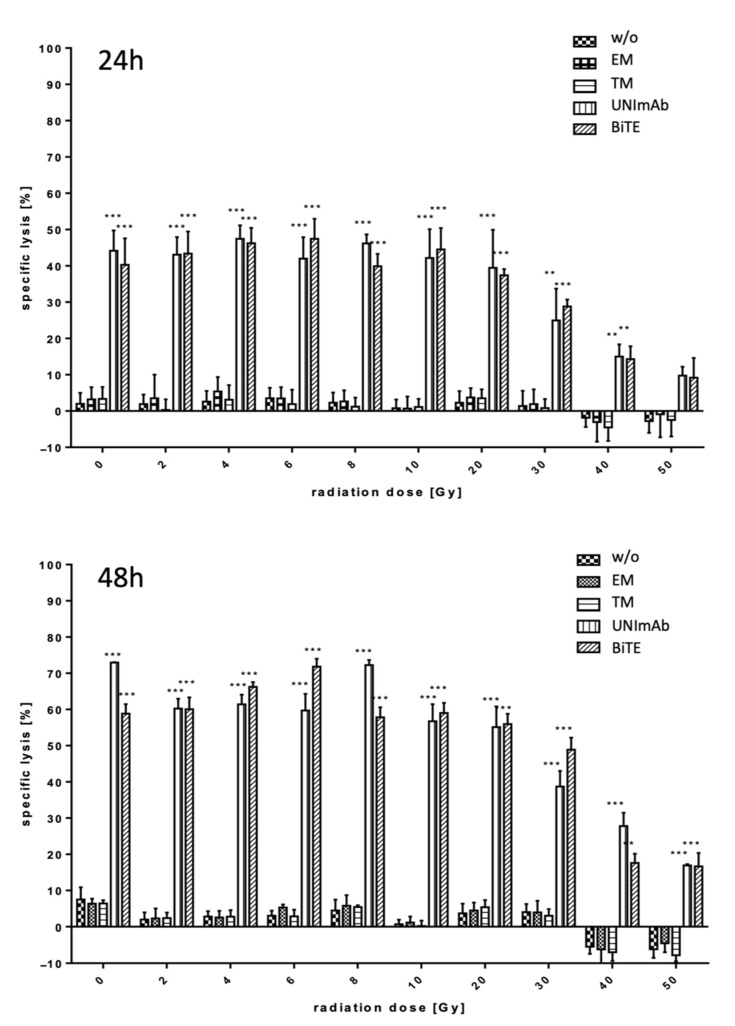
Retargeting of tumor cells with irradiated T cells. Estimation of lysis capability. Pan T cells were redirected either in the absence (w/o) or presence of 30 pmol/mL of the respective Ab components (effector module alone (EM), target module alone (TM), UNImAb complex (UNImAb), conventional BiTE (BiTE)). Both, the CD3-PSCA bsAb (BiTE) or the UNImAb format were able to efficiently lyse the PSCA-positive tumor cells even after radiation with up to 50 Gy (** *p* < 0.01 and *** *p* < 0.001 in relation to the controls (without Ab (w/o), the effector module (EM) or the target module (TM) alone). The data shown were collected for three individual donors. (one-way ANOVA, Bonferroni Multiple Comparison test).

**Figure 3 ijms-23-07922-f003:**
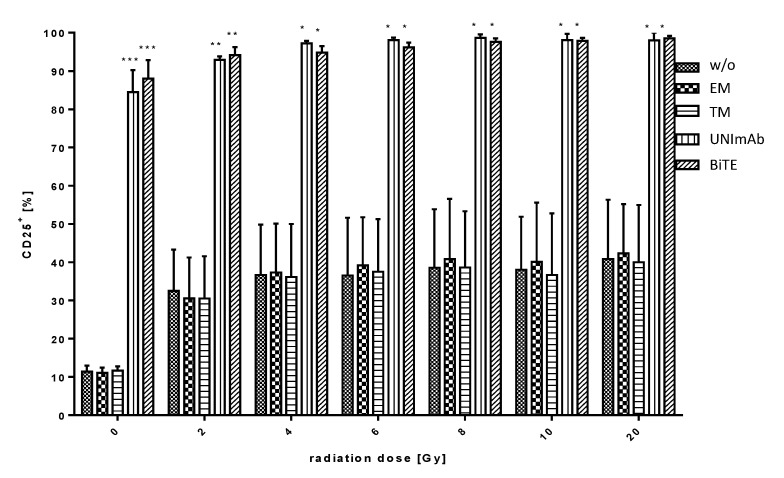
Retargeting of tumor cells with irradiated T cells. Estimation of T cell activation. Pan T cells were irradiated with up to 20 Gy and redirected to PSCA overexpressing PC3 cells either in the absence (w/o) or presence of 30 pmol/mL of the respective Ab components (effector module alone (EM), target module alone (TM), UNImAb complex (UNImAb), conventional BiTE (BiTE)). The pan T cells were incubated with the PC3-PSCA cells for 5 d. Afterwards cells were stained for the expression of the activation marker CD25. The expression level was measured by FACS analysis. Redirection of the T cells resulted in an upregulation of CD25 via both the conventional BiTE and the UNImAb format. Summarized data of three individual donors are shown. (* *p* < 0.05; ** *p* < 0.01 and *** *p* < 0.001 in relation to controls w/o Ab; one-way ANOVA, Bonferroni Multiple Comparison test).

**Figure 4 ijms-23-07922-f004:**
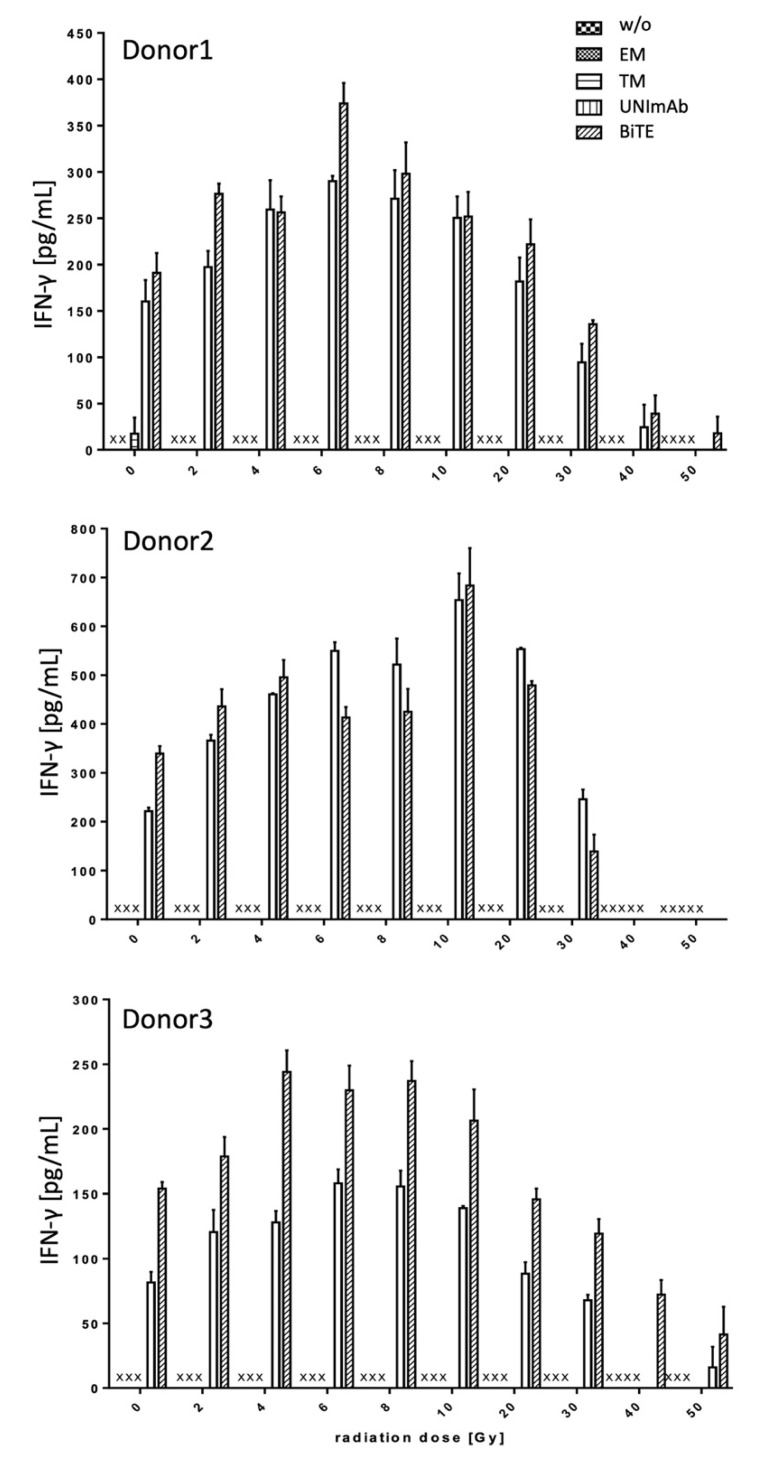
Release of Interferon-γ. The concentrations of Interferon-γ (IFN-γ) were measured in supernatants via ELISA. The mean cytokine concentrations ± SD of triplicates for three different donors are shown. (x… not detectable).

**Figure 5 ijms-23-07922-f005:**
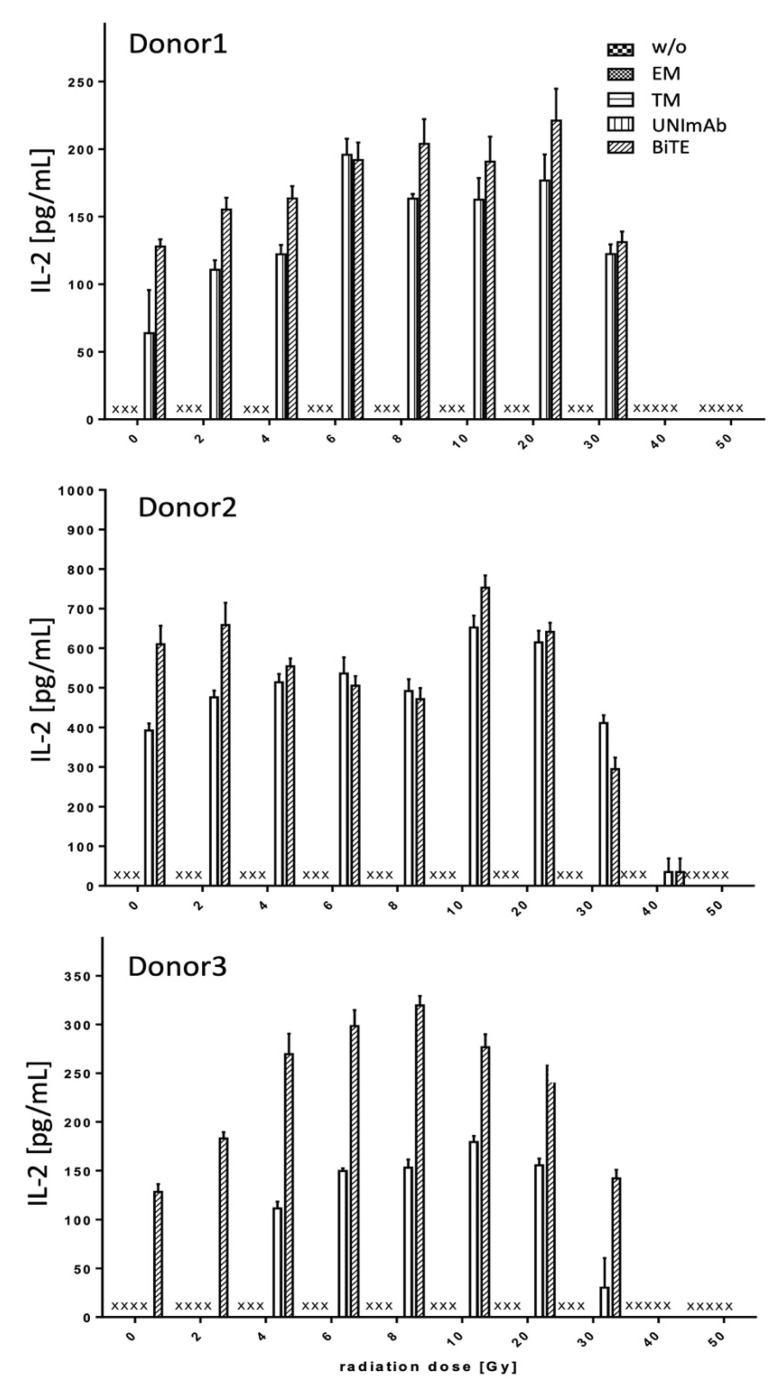
Release of Interleukin 2 (IL-2). The concentrations of IL-2 were measured in supernatants via ELISA. The mean cytokine concentrations ± SD of triplicates for three different donors are shown. (x… not detectable).

**Figure 6 ijms-23-07922-f006:**
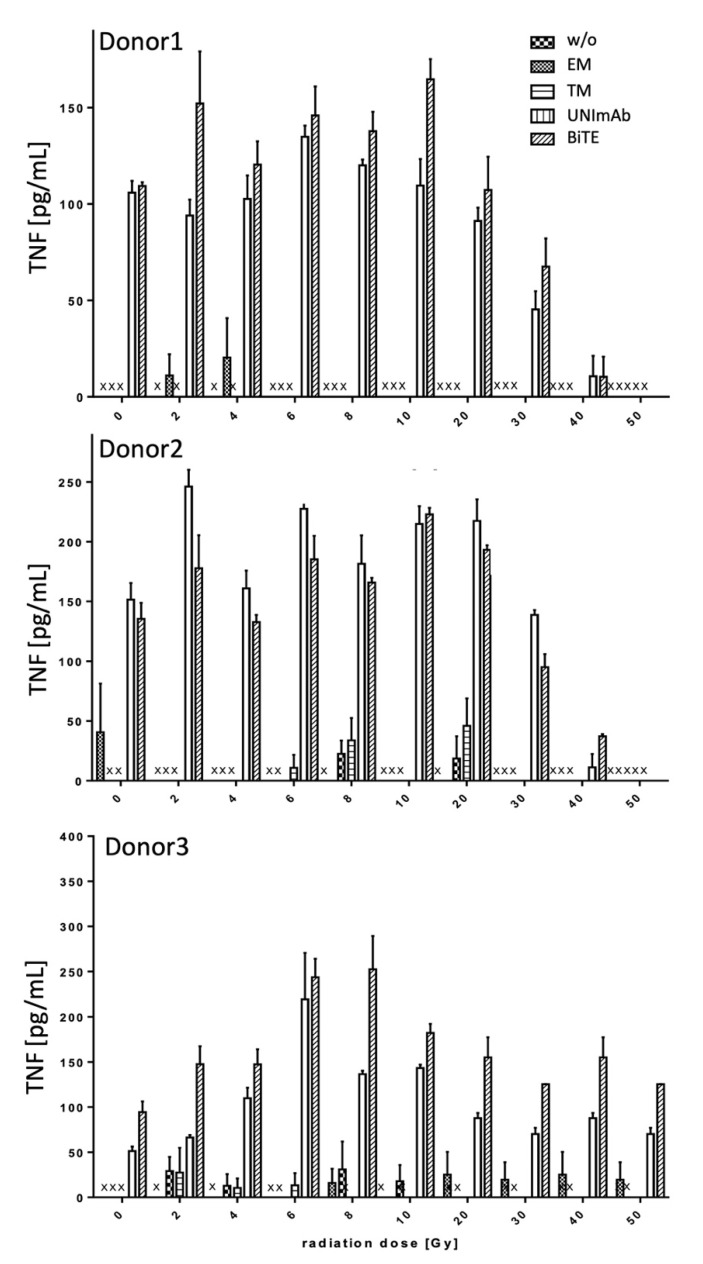
Release of Tumor Necrosis Factor (TNF). The concentrations of TNF were measured in supernatants via ELISA. The mean cytokine concentrations ± SD of triplicates for three different donors are shown. (x… not detectable).

**Figure 7 ijms-23-07922-f007:**
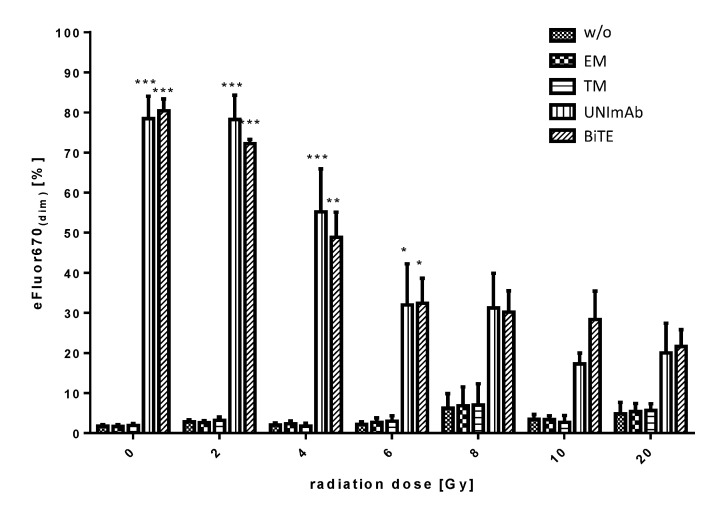
Proliferation of irradiated T cells. Irradiated and unirradiated pan T cells were redirected to PC3-PSCA cells in the absence (w/o) or presence of 30 pmol/mL of the respective Ab components (effector module alone (EM), target module alone (TM), UNImAb complex (UNImAb), conventional BiTE (BiTE)) for 5 d. Before irradiation, the T cells were stained with the proliferation dye eFluor^®^ 670. Proliferation was measured by FACS analysis. Summarized data of three individual donors are shown. (* *p* < 0.05; ** *p* < 0.01 and *** *p* < 0.001 in relation to controls w/o Ab; one-way ANOVA, Bonferroni Multiple Comparison test).

**Figure 8 ijms-23-07922-f008:**
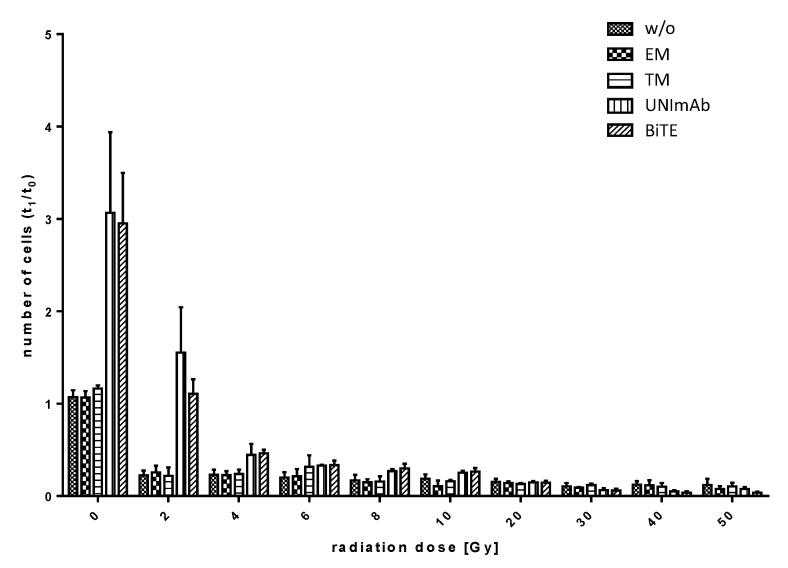
Expansion of irradiated T cells. Irradiated and unirradiated pan T cells were redirected to PC3-PSCA cells in the absence (w/o) or presence of 30 pmol/mL of the respective Ab components (effector module alone (EM), target module alone (TM), UNImAb complex (UNImAb), conventional BiTE (BiTE)) for 5 d. The absolute numbers of T cells were estimated and the ratio (t_1_/t_0_) was calculated. Summarized data of three individual donors are shown.

## Data Availability

The data presented in this study are available on request from the corresponding authors.

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
