# Peer review of "Combining Radiation- with Immunotherapy in Prostate Cancer: Influence of Radiation on T Cells"

_ijms, 2022, doi:10.3390/ijms23147922_

Round 1

Reviewer 1 Report

This manuscript addresses the application of the combination of radiation and immune therapy to treat patients with advanced prostate cancer.  In this study, the authors cocultured PC3 cells with immune cells and recombinant antibodies targeting the prostate stem cell antigen.  T-cell mediated cytotoxicity was measured by 51Cr release.  

The authors tested low dose versus high dose radiation of T-cells to determine if they improve or diminish antitumor responses.  Low dose/fractionate radiation increased the release of IL2 and TNF, whereas 50 Gy diminished T cell function.

There appear to be unmatched parentheses and missing words in 5. Conclusions making it difficult to read and easy to misunderstand.

Author Response

Please note that the lines given in the detailed response are related to the manuscript in the correction mode. In the version, in which all the corrections made are accepted, these line numbers changed.

Reviewer 1:

This manuscript addresses the application of the combination of radiation and immune therapy to treat patients with advanced prostate cancer.  In this study, the authors cocultured PC3 cells with immune cells and recombinant antibodies targeting the prostate stem cell antigen.  T-cell mediated cytotoxicity was measured by 51Cr release.  

The authors tested low dose versus high dose radiation of T-cells to determine if they improve or diminish antitumor responses.  Low dose/fractionate radiation increased the release of IL2 and TNF, whereas 50 Gy diminished T cell function.

There appear to be unmatched parentheses and missing words in 5. Conclusions making it difficult to read and easy to misunderstand.

Response to Reviewer 1:

We are grateful for the overall very positive evaluation of our manuscript by the reviewer and the very careful reading of it. Indeed, we had overlooked the presence of an unmatched parentheses in the section “Conclusions”. We have removed the parentheses which should facilitate the reading and understanding of this section (see page 15, line 379).

Reviewer 2 Report

In their paper on the influence of radiation on T cells  the authors introduce a novel approach in re-targeting T cells against cancer cells from solid tumor, i.e. prostate cancer cells. This novel approach – although only in the preclinical setting – seems to have the advantage that it can be flexibly adapted to various targets on the tumor cells and combined with CARs. I believe that the analysis is of interest to the oncologist in general and specifically to the radio-oncological community. Before publication the following aspects should be addressed.

Title: Although it is obvious to the reader that the results of this analysis can be extended to solid tumors in general, I recommend mentioning the term prostate cancer in the title as the study was performed in prostate cancer cells.

Figure 7. Proliferation of irradiated T cells. If I understand the figure correctly, T cell proliferation starts going down at doses of 4 Gy and above. At the same time, T cells irradiated with 50 Gy still have a significant lysis capability (figure 2).

·         How do these results go together? I assume that the number of proliferating T cells is extremely small after 50 Gy so that the effect of targeting tumors by T cells must be quantitatively extremely low.

·         In radiation treatment courses usually total doses of 60 Gy or more are applied, which lowers the effect even more. In contrast, there are regions further away from the PTV that receive lower doses and – sometimes -  a low dose bath of 2-5 Gy. These areas would be the regions where T cells retain their functionality. How would one counterbalance the effect of “priming T cells in the low dose areas“ on the one hand and destroying them in the high dose areas on the other.

Could the authors please comment on this.

Minor

Lines 49-50: Many solid tumors are primarily treated with chemoradiotherapy and – under specific circumstances – operated. And specifically the concept of re-directing T cells against tumor cells seems – at least for me – most effectively applied in patients who have their tumor in situ. Therefore I would change these lines.

Lines 52-56: I strongly agree that radiotherapy and immunotherapy a “natural partners“ and that these two treatment modalities should be applied simultaneously.

Line 59: leave out “an“ before immunotherapy

Line 232: Nonetheless, in a … for them to do their job - please rephrase the sentence as it does not sound good.

Line 239: Especially of interest may become … - please rephrase the beginning of this sentence.

Author Response

Please note that the lines given in the detailed response are related to the manuscript in the correction mode. In the version, in which all the corrections made are accepted, these line numbers changed.

Reviewer 2:

In their paper on the influence of radiation on T cells  the authors introduce a novel approach in re-targeting T cells against cancer cells from solid tumor, i.e. prostate cancer cells. This novel approach – although only in the preclinical setting – seems to have the advantage that it can be flexibly adapted to various targets on the tumor cells and combined with CARs. I believe that the analysis is of interest to the oncologist in general and specifically to the radio-oncological community. Before publication the following aspects should be addressed.

Response:

We are very grateful for this statement and very kind evaluation of our current work.

Title: Although it is obvious to the reader that the results of this analysis can be extended to solid tumors in general, I recommend mentioning the term prostate cancer in the title as the study was performed in prostate cancer cells.

Response:

To fulfill this request of the reviewer we have changed the title to

Combining Radiation- with Immunotherapy in Prostate Cancer: Influence of Radiation on T cells (see page 1, lines 1 to 3)

Figure 7. Proliferation of irradiated T cells. If I understand the figure correctly, T cell proliferation starts going down at doses of 4 Gy and above. At the same time, T cells irradiated with 50 Gy still have a significant lysis capability (figure 2).

  • How do these results go together? I assume that the number of proliferating T cells is extremely small after 50 Gy so that the effect of targeting tumors by T cells must be quantitatively extremely low.
  • In radiation treatment courses usually total doses of 60 Gy or more are applied, which lowers the effect even more. In contrast, there are regions further away from the PTV that receive lower doses and – sometimes -  a low dose bath of 2-5 Gy. These areas would be the regions where T cells retain their functionality. How would one counterbalance the effect of “priming T cells in the low dose areas“ on the one hand and destroying them in the high dose areas on the other.

Could the authors please comment on this. 

Response:

We are thankful for these three comments. Obviously our message was not sufficiently clear.

To fulfill the request of the reviewer we have rewritten a major portion of the Discussion section (pages 11,12, lanes 238-279). We hope that we could answer all the requested comments with our modifications helping to improve our manuscript as follows: 

Our data show that low radiation doses between 2 and 4 Gy as they are used in fractionated radiation schemes do not harm the killing capability of effector T cells. In contrast, at low doses of irradiation we see an even enhanced lysis of tumor cells although the T lymphocytes appear to be quite sensitive to radiation. In spite of proliferation we see no overall expansion. According to these data, only a portion of the radiated T cells proliferate while another portion of them are dying soon after radiation. The overall loss of T cells is obviously not compensated by proliferation. In order to explain the improved lysis capability, the T cells remaining after radiation should have an enhanced killing capability. Indeed, radiation up to 20 Gy resulted not only in an activation of the immune cells as indicated by the upregulation of CD25, an increased secretion of proinflammatory and T cell maintaining cytokines including IFN-γ, TNF and IL2 but also in an enhancement of their ability to kill their target cells. Bearing in mind that Tregs express high affinity IL2 receptors the increased IL2 levels obviously are not favoring the function of the immunosuppressive Treg cells present in the pan T cell preparation. One possible explanation may be that the Tregs are more sensitive to irradiation. Together with the released proinflammatory cytokines this may be an additional reason for the improved functionality of the effector T cells. Besides fractionated radiation schemes also single radiation doses of up to 60 Gy are clinically used. Interestingly, even when applying such high doses of radiation, a portion of the radiated T cells still remained functional and were even able to proliferate. Taken together, the majority of radiated T cells are going to die but there will be sufficient time for them to attack and kill tumor cells. Moreover, their local activation will convert the cold tumor into a hot one, thereby attracting additional non-irradiated T cells into the tumor tissue. Furthermore, when solid tumors are irradiated there might be a radiation gradient in the tumor and the surrounding healthy tissue. Frequently, T cells are found surrounding solid tumors. These T cells may in part be exposed to favoring low doses which may help to activate them and cause them to enter into the tumor.

In summary, our data indicate that especially low dose of radiation should modulate the TME in favor of a redirection of immune cells but not impair their retargeting capability, thus supporting a combination of RT with targeted immunotherapy. According to our data, one would expect that the immunotherapy should be started in parallel to the RT. An advantage of the combination with RT should not only be true for retargeting via BiTEs but also CAR T cells. We favor the use of modular BiTE and CAR platforms such as the UNImAb or the UniCAR system because their targeting modules can be used as theranostic compound including for imaging, local endoradionuclide therapy, costimulation [49], and immunotherapy [50].

Minor

Lines 49-50: Many solid tumors are primarily treated with chemoradiotherapy and – under specific circumstances – operated. And specifically the concept of re-directing T cells against tumor cells seems – at least for me – most effectively applied in patients who have their tumor in situ. Therefore I would change these lines.

Response:

Our wording may have been slightly misleading. Therefore, we have deleted this sentence and rephrased the following sentence as follows:

Unfortunately, the treatment of patients with cytostatic drugs or radiation causes a selection pressure on the tumor cells favoring the outgrowth of resistant escape variants [e.g. 14-17] (see page 2, lines 55-56).

Lines 52-56: I strongly agree that radiotherapy and immunotherapy a “natural partners“ and that these two treatment modalities should be applied simultaneously.

Response:

We totally agree with the reviewer and are grateful for his or her support.

Line 59: leave out “an“ before immunotherapy

Response:

We have deleted the “an” in line 59. (see page 2, lines 65).

Line 232: Nonetheless, in a … for them to do their job - please rephrase the sentence as it does not sound good.

Response:

In order to respond adequately to the requested comments (see above) we have rephrased most of the Conclusion section. Thereby we also modified the sentence starting originally in line 232 (see now page 13, lines 263-264) as follows:

Taken together, the majority of radiated T cells are going to die but there will be sufficient time for them to attack and kill tumor cells.

Line 239: Especially of interest may become … - please rephrase the beginning of this sentence.

Response:

Again, in order to respond adequately to the requested comments (see above) we have rephrased most of the Discussion section. Thereby we also modified the sentence starting originally in line 239 as follows:

We favor the use of modular BiTE and CAR platforms such as the UNImAb or the UniCAR system because their targeting modules can be used as theranostic compound including for imaging, local endoradionuclide therapy, costimulation [49], and immunotherapy [50] (see page 13, lines 276-277).

 Reviewer 3 Report

Manuscript : IJMS 1804697

Title: Combining Radiation with Immunotherapy: Influence of Radiation on T cells

Date :2022/7/11

Reviewer's report:
This is an interesting manuscript as it’s a comprehensive study on the influence of radiation on T cells from combine radiotherapy and immunotherapy. 
It is well known fact that there’s a synergistic effect of concurrent radiotherapy and immunotherapy. However, little is known on how beam and endoradionuclide associated radiation affects the functionality and the killing capability of tumor infiltrating immune effector cells. This study analyze the effect of radiation on the functionality of immune effector T cells during retargeting of tumor cells. I'm sure this study could help in the decision-making process and guide towards an optimal therapeutic strategy for radioresistance cancer cell.

The MS is well prepared and containing a large amount of data.  Although, there remain some limitation  Nevertheless, it was still well written, thus, it must be published.

Author Response

Reviewer 3

This is an interesting manuscript as it’s a comprehensive study on the influence of radiation on T cells from combine radiotherapy and immunotherapy. It is well known fact that there’s a synergistic effect of concurrent radiotherapy and immunotherapy. However, little is known on how beam and endoradionuclide associated radiation affects the functionality and the killing capability of tumor infiltrating immune effector cells. This study analyze the effect of radiation on the functionality of immune effector T cells during retargeting of tumor cells. I'm sure this study could help in the decision-making process and guide towards an optimal therapeutic strategy for radioresistance cancer cell.

The MS is well prepared and containing a large amount of data.  Although, there remain some limitation  Nevertheless, it was still well written, thus, it must be published.

Response to Reviewer 3:

We are very grateful for the overall very positive evaluation of our manuscript and the recognition of the importance of our work and its publication by the reviewer.